# A Comprehensive Overview of the Developments of Cdc25 Phosphatase Inhibitors

**DOI:** 10.3390/molecules27082389

**Published:** 2022-04-07

**Authors:** Ahmed Bakr Abdelwahab, Eslam Reda El-Sawy, Atef G. Hanna, Denyse Bagrel, Gilbert Kirsch

**Affiliations:** 1Plant Advanced Technologies (PAT), 54500 Vandoeuvre-les-Nancy, France; ahm@plantadvanced.com; 2National Research Centre, Chemistry of Natural Compounds Department, Dokki, Cairo 12622, Egypt; eslamelsawy@gmail.com (E.R.E.-S.); aghanna@hotmail.com (A.G.H.); 3Laboratoire Structure et Réactivité des Systèmes Moléculaires Complexes, UMR CNRS 7565, Université de Lorraine, Campus Bridoux, Rue du Général Delestraint, 57050 Metz, France; denyse.bagrel@univ-lorraine.fr; 4Laboratoire Lorrain de Chimie Moléculaire (L.2.C.M.), Université de Lorraine, 57078 Metz, France

**Keywords:** Cdc25 phosphatases, inhibitor, quinoid, steroid, glycoside, computational technique

## Abstract

Cdc25 phosphatases have been considered promising targets for anticancer development due to the correlation of their overexpression with a wide variety of cancers. In the last two decades, the interest in this subject has considerably increased and many publications have been launched concerning this issue. An overview is constructed based on data analysis of the results of the previous publications covering the years from 1992 to 2021. Thus, the main objective of the current review is to report the chemical structures of Cdc25s inhibitors and answer the question, how to design an inhibitor with better efficacy and lower toxicity?

## 1. Introduction

The cell division cycle 25 (Cdc25) phosphatases are dual-specificity phosphatases (DSPs) that catalyze the dephosphorylation of the Cdk/Cyc protein complex, an important regulator of the human cell cycle [1]. In human cells, three Cdc25 phosphatases are characterized, Cdc25A, Cdc25B, and Cdc25C, which share the similarity of amino acids identity from 20 to 25% for N-terminal and 60% similarity for C-terminal, and are differentially expressed in the cell division cycle [2,3].

Cdc25A is involved in the control of the G1/S transition by dephosphorylating and thus activating Cdk2/cyclin E and cyclin A complexes, as well as controlling the progression into mitosis. Additionally, Cdc25A activates Cdk1/cyclin B provoking the transition G_2_/M [4], while Cdc25B and C mainly regulate the progression at the G2/M transition and mitosis [4,5]. Cdc25B accumulates from the late S and early G2 phases of the cell cycle and its activity peaks at the G2/M transition [4,5]. Cdc25B is proposed to play a “starter” role in triggering mitosis by dephosphorylation and thereby activating Cdk1/cyclin B at the centrosome level [4,5]. On the other hand, Cdc25C is mainly involved in controlling progression into mitosis and this activation is correlated to Cdc25C phosphorylation by Cdk1/cyclinB substrate [5]. The Cdc25 phosphatases also play a crucial role in the checkpoint response that prevents Cdk/cyclin activation following DNA damage [6,7].

Many studies have shown that Cdc25 is highly expressed in cancer. Therefore, it is considered an excellent target for cancer therapy [5,8,9]. Thus, inhibition of these phosphatases may represent a promising therapeutic approach in oncology [10,11,12].

This review is dedicated to searching for safe and efficient Cdc25s inhibitors derived from natural, synthetic, and computational sources. 

## 2. Structure of Cdc25 Phosphatase

The Cdc25 proteins are 450–600 residues in length and divided into two regions, the N-terminal, and C-terminal regions. The N-termini are highly divergent in sequence because of the alternative splicing therein. They contain numerous sites for phosphorylation and ubiquitination involved in regulating phosphatase activity [6]. The C-terminal regions (~60% pairwise identity) contain the catalytic functionality of the Cdc25s. The signature motif (HCX_5_R), containing the catalytic cysteine that defines the active sites of all protein tyrosine phosphatases (PTPs), represents the only significant region of homology between the Cdc25s and other PTPs. Motif C is the catalytic cysteine, while the five X residues form a loop whose backbone amides a hydrogen bond to the phosphate of the substrate. Finally, the motif R is a highly conserved arginine that is required for binding and transition-state stabilization of the phosphate [6].

## 3. Mechanism of Dephosphorylation of Cdc25 Substrate

Like other dual-specificity phosphatases (DSPs), Cdc25s have the ability to hydrolyze phosphoserine/threonine as well as phosphotyrosine residues. Cdc25B-catalyzed dephosphorylation of protein substrates proceeds via a mechanism using a monoprotonated phosphate with an apparent pKa of 6.5. This mechanism is a pH-dependent reaction. Accordingly, at high pH values, the rate of the reaction is slower than at low pH values [13].

The active site contains two key catalytic residues, an unprotonated thiolate and a protonated catalytic acid. The phosphoryl group of the substrate is cradled by the active site loop, forming hydrogen bonds to the backbone amides and the arginine of the HCX_5_R motif. The thiolate anion of the catalytic cysteine lies directly below the phosphoryl group, which sets up an in-line attack of the thiolate on the P-O bond in the first chemical step of the reaction. The departure of the leaving group and formation of the phospho-cysteine intermediate are facilitated by the protonation of the oxyanion by the catalytic acid (still unknown) located in a separate loop. In the second chemical step of the reaction, water serves as a nucleophile in the hydrolysis of the phospho-enzyme intermediate assisted by the aspartate (Figure 1) [6].

It is noteworthy that the thiolate of the active cysteine (Cys473 in Cdc25B) is found to have an extremely high rate of conversion to sulfenic acid. When it is tested with hydrogen peroxide, the result for Cdc25B is 15-fold and 400-fold faster than that for the protein tyrosine phosphatase PTP1B and the cellular reductant glutathione, respectively. This suggests that Cys426 has a preventing role in the formation of sulfenic acid (Cys-SO_2_) in the Cdc25s by forming a rapid intramolecular disulfide linkage with catalytic cysteines (Cys473). This intramolecular disulfide is reversibly and rapidly reduced by cellular thioredoxin/thioredoxin reductase. Thus, the chemistry and kinetics of the active-site cysteines of the Cdc25s support a physiological role for reversible redox-mediated regulation of the Cdc25s as important regulators of the eukaryotic cell cycle [14]. 

## 4. Cdc25 Phosphatases and Cancer

The Cdc25 protein family plays a crucial role in controlling cell proliferation, making it an excellent target for cancer therapy [5,8,9,15]. Cdc25A and Cdc25B are over-expressed in many different primary human cancers [6,16] in the following percentages, respectively: breast cancer: 70% and 57%, thyroid cancer: 17–69% and 36–64%, hepatocellular cancer: 56% and 20%, ovarian cancer: 30% for both of them, colorectal cancer: 47–53% and 43–67%, non-Hodgkin’s lymphoma: 50% and 65%, laryngeal cancer: 41% and 57% and oesophageal cancer: 46–66% and 48–79% [17]. Cdc25B is overexpressed solely in Gastric (78%) endometrial (73%) and prostate (30%) cancers and Gliomas (47%). The incidence of overexpression of Cdc25C has been barely reported [17]. In addition, Cdc25A and Cdc25B are overexpressed together in 30% of tumors while, in 45% of them, one only is overexpressed [17]. 

## 5. Molecular Studies of Cdc25 Phosphatase

The crystal structure of the catalytic domain of Cdc25B displays a well-ordered active site containing a bounded sulfate. This active site loop creates a special environment for cysteine 473 [18,19]. Despite the lack of crystallographic structure in the protein–inhibitor complex and the nonexistence of a deep active site pocket, one of the largest cavities on the surface of Cdc25B adjacent 0.6 Å to the active site reveals good binding envelop for the small inhibitory molecules during docking processes (Figure 2) [18]. The residues that surround the inhibitor binding pocket adjacent to the active site contain Arg482, Arg544, and Thr547 besides other residues [19]. 

## 6. Interaction between Cdc25 Phosphatase and Its Protein Substrate

Several investigations have tried to predict a docked orientation for Cdc25B with its Cdk2-pTpY-CycA protein substrate using different docking techniques [20]. The available crystal structure containing a bound sulfate (1QB0) at the active site is expected to be a good representative of the phosphate group of the phosphorylated protein substrate [21]. However, on the Cdk2-pTpY-CycA substrate, the exact orientation of the *β*-hairpin loop encompassing the adjacent Thr14 and Tyr15 sites of phosphorylation is unknown. The second point of contact between Cdc25B and its protein substrate is hot-spot residues (Arg488 and Tyr497). They are >20 Å from the active site and expected to make a contact with either Cdk2 or cyclin A at the binding interface [20].

Computational docking of Cdk2-pTpY-CycA to the enzyme followed by refinement using molecular dynamics was performed. In addition, validation of the docking using X-ray crystallography was applied to the crystal structure of the substrate-trapping mutant of Cdc25B [22,23]. The phosphate of pThr14 is cradled within the active site loop with numerous hydrogen bonds to the amides backbone including the side chain of Arg479 of the CX5R signature motif. This phosphate is additionally coordinated in the active site pocket by contributions from the amide backbone of Thr14 and the side chain of Arg36 from the substrate [20].

Further studies have been conducted to clarify the exact model for enzyme–substrate interaction. Some of these studies used small molecules such as *p*-nitrophenyl phosphate (*p*NPP) as substrate and applied the recently developed QM/MM Minimum Free Energy Path method. This was intended to gain an understanding of the dephosphorylation mechanism [1].

## 7. Cdc25 Inhibitors

A diverse number of structural inhibitors of the Cdc25 phosphatases have been identified and reported in the literature. Many of the Cdc25 inhibitors have been discovered via either the isolation of new scaffolds by High Throughput Screening (HTS) or optimization of pre-existing inhibitor scaffolds [24]. 

### 7.1. Cdc25 Inhibitors Based on the Quinoid Structure

Menadione (vitamin K_3_) (**1**) is one of the first *para* quinoid compounds that proved its activity against Cdc25 A, B, and C (*Ki*: 38 ± 4, 95 ± 3, and 20 ± 4 µM, respectively). Suggestions to modify the structure by delivering polar derivatives of this nucleus did not lead to increased activity in contrast to what was expected [10,11]. 

Cpd-5, [2-(2-mercaptoethanol)-3-methyl-1,4-naphthoquinone] (**2**) is a VK_3_ analogue and had selective and irreversible Cdc25 inhibition. It arrested cell division at G1-S and G2-M checkpoints. Its selectivity of inhibition of recombinant human Cdc25B2 was distinctive with minor or no effect upon VHR and PTP1B [25] with IC_50_ values of 5, 2, and 2 µM on Cdc25A, B, C, respectively [26,27].

The fluorinated analog **3** of Cpd-5 (**2**) had advantages over its parent nucleus since it was less toxic. The inductive effect of fluorine atom resulted in the liberation of lower ROS amount, thus lowering the toxicity. Meanwhile, it had higher potency with IC_50_ of 0.8, 1, and 50 µM against Cdc25A, B, and C, respectively [27]. 

Another correspondence to vitamin K is Cpd-42 (**4**) and its isomer **5.** Both had the same IC_50_ toward Cdc25A (1.5 µM), which may explain the role of these compounds in the suppression of a Hep3B hepatoma cell line [28].

New biotin-containing Cpd-5, derivative **6**, and its isomer **7**, were found to be Cdc25 inhibitors by the arylation of the active cysteine of Cdc25A, B, and C. Cpd-5 derivative **6** and its isomer **7** exhibited a selective manner of inhibitions with IC_50_ of 1.5, 30, and 100 µM for Cdc25A, VHR, and PTB1B, respectively. Carbonyl oxygens of the derivative appeared to interact with both Arg482 and Arg544 near the catalytic cysteine 473 [29]. 

A quinone-based small molecule compound NSC 95397 (**8**) was the most active hit among many structurally modified vitamin K3 [30]. Dihydroxy NSC-95397 (**9**), a hydroxyl derivative of **8**, designed by Peyregne et al., was assumed to be more bioactive with IC_50_ of 1.2 ± 0.2, 1.5 ± 0.5, 1.3 ± 0.2, and 1.8 ± 0.5 µM, toward Hep-3B, HepG2, PLC/PRL/5 hepatoma cells and MCF7 breast cancer cells, respectively [26]. The fluorinated analog **10** of NSC 95397 **8** had no effect [27]. 

On the other hand, the naphthalene-type analog of vitamin K_3_ **11** had Cdc25A phosphatase-inhibitory activity equal to IC_50_ 0.4 µM [31]. In addition, naphthoquinone derivatives, containing malonic **12** or carboxylic acids **13** groups were introduced to mimic the role of phosphate moieties of Cdk complexes. The malonic acid derivative **12** did not show the expected rise of the activity with IC_50_ value 10.7 ± 0.5 µM on Cdc25B and displayed moderate cytotoxicities against the HeLa cell line [32], whereas compound **13** exhibited IC_50_ of 4.55 ± 0.16 µM toward Cdc25B and weak activity against HeLa cell lines [32]. Bis(methoxycarbonylmethylthio)heteroquinone analog of vitamin K_3_ **14** showed reasonable activity against Cdc25B with IC_50_ of 1.17 ± 0.04 μM [33]. 

A novel coumarin-quinone derivative SV37 **15** was synthesized in a trial to decrease the toxicity and inhibit Cdc25 phosphatases. Despite being less active than a sole quinone moiety on the same biological targets, it showed IC_50_ effects on the Cdc25B and C catalytic domain of 45.5 ± 8 μM. It also exhibited IC_50_ of 0.9 ± 0, 9.5 ± 1.0, 18.1 ± 1.3, and 11.2 ± 0.7 µM against Hep-3B, HTERT-HME1, Hela, and Fem-X, respectively [34]. 

The reversibility of the aforementioned compound was tested by the mass spectrometry (MALDI–TOFMS) technique and was proved to reversibly interact with Cdc25 A and C (Figure 3, Table 1) [35].

A new family of potent inhibitors **16**–**28** of the dual-specificity protein phosphatase Cdc25 of quinone structure and analogs of vitamin K_3_ was reported (Figure 4, Table 2). 6-Chloro-7-(2-morpholin-4-ylethylamino) quinoline-5,8-dione (NSC 663284) (**16**) was described as a very potent irreversible inhibitor of Cdc25 A, B with IC_50_: 210 nM; additionally, **16** showed 20- and 450-fold selectivity for Cdc25B compared to VHR or PTP1B phosphatases, respectively [36]. The NSC 663284 (**16**) showed a direct binding to one of the two serine residues in the active site loop of the Cdc25A catalytic domain (Ser114, corresponding to Ser434 in full-length Cdc25A) [36]. Similar compounds JUN 1111 (**17**) also acted by induction of ROS, which led to irreversible oxidation of the catalytic cysteine thiol to sulfonic acid; therefore, cells were arrested in G1 and G2/M phases of the cell cycle. Enzyme suppression was found to be affected by medium pH, in the presence of catalase and reductants (dithiothreitol and glutathione) [37].

Quinone structures **18**, **19**, **20** were reported to have IC_50_ toward Cdc25B of 0.5 ± 0.1, 1.1 ± 0.1, and 5.3 ± 0.6 µM, respectively. They were designed to have half-wave potentials less than 105 mV, to obtain a stable molecule in its reduced state. Theoretically, it would produce a controlled amount of ROS or H_2_O_2_. Unfortunately, it was found that their redox cycle and E1/2 did not match what was expected [38]. 

Caulibugulones A-F (**21**–**26**) are isoquinoline derivatives obtained from the marine bryozoan *Caulibugula intermis* exhibiting a IC_50_ values range of 2.7–32.5 μM towards Cdc25B in very intensive selectivity compared with other phosphatases VHR and PTP1B [39]. Caulibugulones A-F (**21**–**26**), displayed distinct patterns of differential cytotoxicity against 60 antitumor cell lines. The results showed that **21**–**26** revealed in vitro cytotoxicity with IC_50_ values in the range of 0.03–1.67 µg/mL against murine tumor cell line IC-2^WT^ [40,41]. By investigating the inhibitory mechanism of caulibugulone A (**21)**, it was found that its inhibition effect was not attributed to the release of ROS, as caulibugulone A (21) releases a smaller amount of ROS. The inhibition appeared to take place by the stress-activated protein kinase p38 pathway. This protein controls mitosis under the condition of osmotic pressure by degradation of Cdc25A through its phosphorylation at Ser75. Caulibugulone A (**21**) activated p38 which leads to degradation of Cdc25A with little or no loss of either Cdc25B or C [41]. 

Dimethyl 2,2′-((5,8-dioxo-5,8-dihydroquinoline-6,7-diyl)bis(sulfanediyl)) diacetate (**27**) showed IC_50_ of 1.63 ± 0.21 μM on Cdc25B. In addition, **27** showed inhibition percent of 79 ± 0.9 and 77 ± 2.1% at 100 μM toward HeLa and MiaPaCa-2 cell lines, respectively, while at 10 μM it showed an inhibition of 13 ± 0.9 and 24 ± 15.4% against the same cell lines, respectively [33].

In 2021, Zhan and his coworkers utilized miniaturized synthesis to generate several quinonoid-focused libraries, by standard CuAAC reaction and HBTU-based amide coupling chemistry. They found that the naphthoquinone (M5N36, **28**) exhibited higher inhibitory activity (Cdc25A: IC_50_ = 0.15 ± 0.05 μM, Cdc25B: IC_50_ = 0.19 ± 0.06 μM, and Cdc25C: IC_50_ = 0.06 ± 0.06 μM). Hence, M5N36 displayed about three-fold higher potentency against Cdc25C than Cdc25A and B. Thus, M5N36 was recognized to be a promising novel Cdc25C inhibitor [42]. 

**Table 2 molecules-27-02389-t002:** Cdc25 inhibitors based on quinoid structure analogs of vitamin K3 substituted with heteroatoms in the benzene ring.

ID	Cdc25 Inhibitory			A549	IC-2^WT^	HeLa	MiaPaCa-2	MDA-MB-436	PC-3
	A	B	C						
**16** [36]	0.5 ± 0.01 *	0.21 ± 0.08 *	0.47 ± 0.09 *	1.48 ± 0.04				0.5	1
		0.91 ± 0.36 *							
**17** [37]	0.38 ± 0.06	1.8 ± 0.7	0.66 ± 0.27					0.1	0.8
**18** [38]		0.5 ± 0.1		22.28 ± 0.62					
**19** [38]		1.1 ± 0.1		2.69 ± 0.08					
**20** [38]		5.3 ± 0.6		9.52 ± 0.33					
**21** [40,41]	3.38 ± 0.62	6.7 ± 1.3	5.43 ± 1.22		0.34				
**22** [40,41]	1.51 ± 0.2	2.7 ± 0.5	2.68 ± 0.23		0.22				
**23** [40,41]	2.57 ± 0.60	5.4 ± 0.7	3.31 ± 0.28		0.28				
**24** [40,41]	4.89 ± 0.82	19.1 ± 0.3	10.8 ± 0.46		1.67				
**25** [40,41]	18.2 ± 1.13	32.5 ± 3.6	16.6 ± 0.98		0.03				
**26** [40,41]					0.01				
**27** [33]						13 ± 0.9 ** (10 µM)	24 ± 15.4 ** (10 µM)		
						79 ± 0.9 ** (100 µM)	77 ± 2.1 ** (100 µM)		
**28** [42]	0.15 ± 0.05	0.19 ± 0.06	0.06 ± 0.06						

* Inhibition of cells line measured by nM; ** inhibition of cells line measured by % of inhibition. Other than that, all IC_50_ values measured by µM.

New quinone structures based on tanshinones were reported as potent inhibitors of Cdc25 (Figure 5, Table 3). Compound **29** was synthesized on a pattern of tanshinone and revealed activity toward Cdc25B (IC_50_ = 3.21 ± 0.76 µM) [43].

Additionally, two other series of tanshinone derivatives **30**–**32** (Figure 5) were synthesized by replacement of the methyl group in site 3 with different ester, acetyl, or alcohol as withdrawing groups. It was found that these electron-withdrawing groups decrease significantly the antitumor activity while the modification at position 2 of the structure did not change the activity. The IC_50_ of **30**, **31**, and **32** toward Cdc25A and B were 0.73 ± 0.1 and 0.65 ± 0.09, 0.73 ± 0.04 and 0.48 ± 0.03, 1.10 ± 0.11 and 0.89 ± 0.07 µM, respectively. They were also tested on A549 tumor cell lines and showed IC_50_ of 1.66 ± 0.13, 3.14 ± 0.03, and 2.80 ± 0.2 µM, respectively [44]. The 3-benzoyl substitution in 3-benzoyl-naphtho[1,2-*b*]furan-4,5-dione (5169131) (**33**) stimulated selective inhibition of the Cdc25 enzyme (IC_50_: 5–10.5 µM) against the three isoforms of the enzyme [45] (Figure 5).

Nocardines A (**34**) is a natural tanshinone isolated from *Nocardia* (strain TP-A0248). It exhibited IC_50_ of 17 µM against Cdc25B and was slightly more potent toward PTP1B (IC_50_ 14 µM). On the other hand, Nocardines A (**33**) revealed good cytotoxic activity on HeLa cells (IC_50_ = 0.38 µM) and SBC-5 (IC_50_ = 0.54 µM). It also affected U937 human myeloid leukemia cell lines with IC_50_ of 6.25 µM by inducing apoptosis [46] (Figure 5).

Huang et al., synthesized new miltirones as quinones analogs and evaluated their inhibitory activity against Cdc25. Compound **35** of these analogs was the most potent with IC_50_ of 3.96 ± 0.63, 3.16 ± 0.22, and 4.11 ± 0.40 µM against Cdc25A, B, and C, respectively [47] (Figure 5).

Lazo et al., constructed a library concerning small molecules structurally related to paraquinoid and non-para quinoid to be Cdc25 inhibitors. Four of them were the most potent compounds NSC 135880 (**36**), 139049 (**37**) and 115447 (**38**), with IC_50_ = 0.125 ± 0.036, 0.210 ± 0.071, and 0.450 ± 0.130 µM, respectively, against Cdc25B phosphatase [48] (Figure 6).

Compound **39** was discovered as a Cdc25 inhibitor during a biological evaluation of the Dnacin (antitumor antibiotic) analogs. At a concentration of 100 µM, it showed considerable activity against Cdc25A and B (IC_50_ = 91.4 ± 4.3 and 99.5% ± 1.4) and moderate activity toward VHR (IC_50_ = 67.2 ± 6.3) and PTP1B (IC_50_ = 48 ± 4.1) [49].

One of the earlier Cdc25 inhibitors of quinoid scaffold, which naturally originated from *Streptomyces* sp., is **40** which showed inhibition activity against Cdc25A with IC_50_ of 15.5% at 10 µM, recording the most potent activities among the isolated compounds [50] (Figure 6).

The most powerful quinone that has ever been discovered is adociaquinones B (**41**) which was separated from *Xestospongia* sp. with IC_50_ of 0.08 ± 0.01, 0.07 ± 0.01 µM against Cdc25A and the B phosphatase catalytic domain [51] (Figure 6). Simplified adociaquinone B analogs **42** and **43** were synthesized and showed IC_50_ of 0.94 and 0.88 µM against Cdc25B, respectively. When they were tested against human ovarian cell line A2780 they showed IC_50_ of 4.3 ± 2.3, 2.1 ± 0.02 µM, respectively (Figure 6). They have an advantage over adociaquinone B in their smaller size, which makes them more accessible to the binding site and also gives versatility for derivatization [52].

BN82685 **44**, an additional quinone-based Cdc25 inhibitor, showed irreversible inhibition against Cdc25 A, B, and C with comparable potency of IC_50_ of 250, 250, and 170 µM, respectively. BN82685 **44** showed activity against *xenografted* tumors upon testing in vivo and was reported as one of the few inhibitors of Cdc25 by oral administration [53] (Figure 7). All the cancer cell lines were sensitive to BN82685 treatment ranging from 250 to 1 µmol/L. The most sensitive were DU-145, Mia PaCa-2, and A2058 (IC_50_ of 90, 118, and 134 µmol/L, respectively), while the lowest sensitivity was for IMR-90 normal fibroblasts with IC_50_ of 1 µmol/L [53].

A bis quinone IRC-083864 (**45**) inhibitor of Cdc25 phosphatases was reported. It displayed IC_50_ of 23 ± 0.4, 11 ± 1.5, 23 ± 1.7, 26 ± 4.4, and 53 ± 9.4 nM against Cdc25C, Cdc25C-catalytic domain, Cdc25A, CdcB2, CdcB3, respectively [54] (Figure 6). Additionally, it showed a potent inhibitory activity on different cell lines including MIA PaCa-2, DU145, LNCaP, MCF-7, and A2058 in nanomolar ranges. Additionally, it showed promising results against several cell lines that reported resistance toward some well-known active agents [54]. The new quinone-based tetracycle **46** provided activity toward Cdc25s with IC_50_: 5.8, 14.4 and 2.3 μM against Cdc25A, B, and C, respectively [55].

Although **47** and **48** are not quinoids, they were inspired by IRC-083864 **46** (Figure 6) [56]. Compound **47** showed IC_50_ values against Cdc25B (3.4 ± 1.1), and PTP1 (30.6 ± 11.3) at 100 µM, and its inhibition mechanism may be related to ROS production [56]. In contrast, **48** was not considered ROS dependent; meanwhile, it showed inhibitor activity against Cdc25B, PTP1and VHR with IC_50_ of 1.38 ± 0.1, 24.2 ± 4.5, and 31% at 10 µM, respectively [56].

Compound **49** was intended to be applied as the precursor of **50**, but surprisingly it was found to be much more active than the compound of focus (Figure 6). Interestingly, **49** achieved inhibition against Cdc25A (IC_50_= 2.1 µM, 25-fold higher potency than B and C isoforms). It also induced apoptosis in human hepatoma cell line Hep3B with IC_50_ of 2.5 µM representing 2.5-fold higher potency than Cpd-5, and 3.2-fold higher potency than Vitamin K_3_ without harming the normal cells as it did not generate ROS [57,58]. Non-quinone Cpd 5 analog **50** was synthesized in order to obtain a non-quinoid structure to produce less or no ROS. It exhibited an IC_50_ value of 7.5 µM against Cdc25A.

The indolyldihydroxyquinones group was synthesized as isosteric analogs of paraquinoid. The best derivative was **51** with reversible inhibitory activity (IC_50_ of 1.0, 1.0, and 3.5 µM) against Cdc25 A, B, and C, respectively [59] (Figure 6).

Quniolinone **52** may be considered as an isosteric analog to quinolinequinone with partial quinoid character (Figure 6). Quniolinone **52** Was produced after several synthetic steps and inhibited Cdc25 phosphatase at 5 µM. Additionally, it was bioassayed with different cell lines but did not appear to have potential activity [60].

### 7.2. Cdc25 Inhibitors Based on Steroid and Dysidolide-like Compounds 

Dysidiolide **53** was the first naturally discovered Cdc25A phosphatase inhibitor (IC_50_ 9.4 µM) with antitumor activity [61]. The *γ*-hydroxybutenolide group may surrogate the phosphate moiety of the substrate. The hydrophobic side chain is suggested to fill the hydrophobic groove adjacent to the active cysteine [62]. On contrary, Blanchard et al., suggested that the activity of dysidiolide was related to the other component in the crude extract not to the dysidiolide itself [63,64]. The IC50’s of their synthetic dysidiolide were high (35 and 87 µM towards Cdc25A and Cdc25B, respectively); additionally, it showed an inhibitor activity of 5.4 and 7.1 µM against SBC-5 and HL60 cell lines [64].

Dysidiolide **53** was used as a base structure for further optimization. The dysidiolide analog **54** was one of the first examples of **53** analogs with IC_50_ of 2.2 µM toward Cdc25A phosphatase [62,65,66]. *Epi*-dysidiolid **55** and **56** are isomers of dysidiolid, and showed certain activity toward Cdc25s. Briefly, **55** showed IC_50_ of 15 and 19 µM for Cdc25B, whereas **55** revealed IC_50_ of 13 and 18 µM against Cdc25A and B, respectively. Additionally, compounds **55** and **56** exhibited inhibitory activity against SBC-5 and HL60 cancer cell lines with IC_50_ of 16 and 13.1 µM for **55**, and 1.3 and 1.0 µM for **56** [64].

Compound 57 is a product of ozonolysis of α,β-unsaturated keto acid, 5,6-seco-5-oxo-3-cholesten-6-oic acid, exhibiting moderate inhibitory activities against human Cdc25A protein phosphatase with IC_50_ equal to 7.9 µM [67]. On the other hand, 58 was synthesized by a solid phase technique and showed Cdc25A phosphatase inhibition with IC_50_ of 0.8 µM [68].

Depending on the fact that dysidiolid contains two parts, one hydrophobic and other hydrophilic, some isosteres were synthesized **59**–**64** [69,70,71,72,73,74]. Attachment of the hydrophilic moiety of dysidoiolid to vitamin D_3_ scaffold led to a group of analogs **59**–**61**. Compound **59** recorded an activity towards Cdc25A and SBC-5 cell lines with IC_50_ of 7.7 and 47 µM, respectively. Additionally, it arrested the G1 phase of the HL60 tumor cell line [69]. Furthermore, **60** and **61** showed inhibitory activity towards Cdc25A with IC_50_ of 0.48 and 0.7 µM, besides 1.44 and 3.7 µM towards Cdc25B, sequentially [70,71]. Compound **62** was designed by delivering a long aliphatic chain and showed IC_50_ on Cdc25A of 1.9 ± 1.0 and 1.6 ± 0.2 µM for *E* and *Z* isomers, respectively [72], whereas **63** and **64** are combinations of malic and oleic acid moieties. Compound **63** inhibited Cdc25s A, B, and C by IC_50_ of 5.1 ± 0.3, 3.6 ± 0.4, and 4.3 ± 1.1 µM, respectively, while **64** showed IC_50_ of 4.5 ± 1.7, 6.7 ± 0.6, and 8.8 ± 1.3 µM against Cdc25s A, B, and C. They induced apoptosis in the breast cancer (MCF7) and vincristine-resistant (Vcr-R) cell lines with no influence of ROS. They also synergetically augmented the efficiency of Adriamycin and cisplatin in four mentioned cell lines [73,74].

Coscinosulfate **65** was extracted from marine sponge *Coscinoderma matthewsi* and showed the best activity among its synthetic analogs on Cdc25A phosphatase with IC_50_ of 3 µM [75].

2-Methoxyestradiol (2-ME) **66** suppressed the growth of hepatoma cells by IC_50_ of 0.5–3 µM without affecting the normal hepatic cell of the rat up to a concentration of 20 µM. The inhibition mechanism was suggested to happen by competitive inhibition of Cdc25s on the catalytic cysteine using biotin-labeled Cpd 5 (Cpd 5-Bt) as a ligand. The measured IC_50_ 0f **65** was 1, 1, and 10 µM for Cdc25A, B, and C [76].

The Chemical structures of compounds **53**–**66** and their biological activity are illustrated in Figure 7 and Table 4. 

### 7.3. Cdc25 Inhibitors Based on Glycosidic Structures

Some glycosidic derivatives were proved to be potential inhibitors (Figure 8). Compound **67** was one of the early examples shown by Yang et al., with medium selectivity on Cdc25B with IC_50_ of 3.9 µM, and on PTP1B with IC_50_ of 8.3 µM compared with other phosphatases [77]. Compound **68** was another inhibitor of this group with IC_50_ of 6.2 ± 3.0 µM on Cdc25B. Furthermore, **68** was potent against A549, Hela, and HCT-116 cell lines with IC_50_ of 35.2 ± 3.2, 25.3 ± 2.4, and 12.8 ± 1.3 µM, respectively [78]. The bis-triazole **69** was synthesized and showed moderate activity on Cdc25B with IC_50_ of 11.1 ± 1.4 μM and inhibited PTP1B by IC_50_ = 25 μM [79].

### 7.4. Miscellaneous Cdc25 Inhibitors 

The synthetic approach of RK-682, isolated from *Streptomyces sp*., led to compound **70** which is an *R* stereoisomer that inhibited Cdc25A and B with equal potency of 34 µM. It was more potent against VHR with IC_50_ of 3.4 µM and not active against protein serine/threonine phosphatase l (PP1) [80]. Another example is compound **71** which showed activity with IC_50_ of 0.38 ± 0.02 µM against Cdc25B [81].

To replace the resonating negative core of RK-682 with a neutral one as a suggestion to increase cell permeability and selectivity, the enamine **72** (RE44) was developed. It revealed suppression of Cdc25A (IC_50_: 13.5 ± 3.4 μM) and Cdc25B (IC_50_:4.26 ± 0.17 μM). No production of ROS was recorded [82].

Depending on the combinatorial chemistry study of some natural products, known as phosphatase inhibitors, a pharmacophore model was postulated and a resulting structure was identified [83]. The most active compound of this template was SC-ααδ9 **73** with IC_50_ of 15 µM against Cdc25A and B [84].

Sulfonylated aminothiazole derivative **74** was synthesized as an analog of SC-ααδ9 **72** with a smaller molecular weight and low selectivity toward Cdc25B (IC_50_: 26 ± 4 µM) in comparison with VHR (IC_50_: 32 ± 9 µM) and PTP1B (IC_50_: 24± 4 µM) [85].

FY21-αα09 **75** is a modified form of SC-ααδ9 **73**. FY21-αα09 **75** inhibited the Cdc25B2 at concentrations of 3 and 100 µM with inhibition percentages of 36 ± 1.3, and 83 ± 0.6%, respectively. Moreover, at the same concentration (3 and 100 µM) the FY21-αα09 **75** showed inhibition percentages towards VHR and PTP1B with values of 11 ± 2.7, 95 ± 2.6, 3.4 ± 1.6, and 78 ± 1.7, respectively [86].

In a patent released in 2002 by Jill et al., an intensive effort was exerted to identify peptidic inhibitors. They combined X-ray and computational methods to identify the 3-D image of amino acids in the natural substrate of the Cdc25 enzyme. Combinatorial chemistry was utilized to synthesize a wide variety of peptidic compounds comprising the crucial amino acids moieties defined in the substrate as reliable for binding with the Cdc25 enzyme. Compound **76** Is a pentapeptide invented by the above-mentioned technique with IC_50_ of 1.1 μM against Cdc25A [87].

Compound **77** is a thiazolidinone derivative. Its activity as a selective Cdc25A inhibitor was reported in a comparative analysis with MKP-1 by stopping the cell cycle in Go/Gi or S phases. On the other hand, **77** provided cytotoxic activity against different human cancer cell lines including MDA-MB-435, PC-3, A549, and Bx-PC3 [88].

Macrocyclic inhibitors are structurally derived from the steroid and postulated to have the same activity as the steroid. This group showed good selectivity in comparison with other phosphatases. Compound **78** was the best among this group, as it showed inhibition activity with IC_50_ of 3.5 µM towards CdcB25 phosphatase [89].

BN82002 **79** has been synthesized and introduced as a potent Cdc25C phosphatase with IC_50_ on Cdc25A, B2, B3, C, and C-cat equal to 2.4 ± 1.2, 3.9 ± 0.01, 6.3 ± 4.1, 5.4 ± 3.4, and 4.6 ± 0.5 µM, respectively. When **79** was tested on different cell lines, it gave the following results of IC_50_ towards Mia PaCa-2 (7.2 ± 0.9 µM), DU-145 (13.5 ± 0.8 µM), U-87 MG (13.9 ± 3.5 µM), LNCaP (22.4 ± 3.1 µM), HT-29 (32.6 ± 8.9 µM), U2OS (11.2 ± 0.6 µM), and HeLa (23 ± 4.5 µM). One of BN82002 derivatives is compound **80**, which showed IC_50_ of 3.9 ± 0.4 µM against Cdc25C during in vitro enzyme assay [90,91].

Poly prenyl furanes 81 and hydroquinones 82 were isolated from three sponge species and demonstrated potential activity towards Cdc25A phosphatase with IC_50_ of 2.5 and 0.4 µM [92].

Suramin **83**, which was originally used to treat sleeping sickness, revealed IC_50_ towards Cdc25A of 1.5 ± 0.2 µM. In 2004, McCain et al., synthesized NF171 **84**, NF279 **85**, and NF280 **86** that were closely similar to suramin and showed an inhibitory activity with IC_50_’s of 1.8 ± 0.4, 1.0 ± 0.3, and 0.84 ± 0.28 µM towards Cdc25A [93,94].

Compounds **87** and **88** were structurally inspired by suramin. Compound **87** showed high potency with IC_50_ equal to 0.38 ± 0.04 on Cdc25A but with no selectivity, whereas compound **88** showed IC_50_ of 3.2 ± 0.3 on the same target but with great selectivity versus other phosphatases [94].

Fascaplysin **89** is an alkaloid isolated from *Thorectandra sp*. sponge and revealed an IC_50_ value of 1.0 µg/mL against Cdc25B. Sesterterpenoids **90** was obtained from the previously mentioned organism and demonstrated IC_50_ (µg/mL) of 1.6 on Cdc25B [95].

Diterpenoid **91** was obtained from a sea anemone and revealed an inhibitory activity against Cdc25B with IC_50_ of 1.6 µg/mL [96].

Some dimers of cinnamaldehyde were synthesized by Shin et al.; compound **92**, a member of this group, inhibited Cdc25B by 95% at 10 µM with IC_50_ equal to 1.94 µM, whereas it showed less activity on Cdc25A with IC_50_ of 19.24 µM [97].

Cheon et al., invented a large number of thiazolidine derivatives, and they evaluated the efficiency of one of them (**93**). Compound **93** was examined against Cdc25B and some cancerous cell lines, namely A-549, HT29, and MCF-7. The IC_50_ of this compound, when incubated with the protein, was 2.26 µM and it appeared to be more active against the previously mentioned cell lines in comparison with the reference at 10 µM with IC_50_ of 67.5, 59.7, and 67.3%, respectively [98].

LGH00045 **94** was discovered by high-throughput screening and revealed good selectivity against Cdc25 phosphatases. The IC_50_ values of **94** against Cdc25A and Cdc25B were 0.53 ± 0.02 and 0.82 ± 0.08 μM, respectively [99].

Wook Ham and his coworkers identified eight novel Cdc25B inhibitors by virtual screening. They found that compounds **95**, **96** reversibly inhibited Cdc25B by IC_50_ values of 0.21 ± 0.06 µM and 0.20 ± 0.09 µM, respectively. Additionally, compounds **95**, and **96** revealed 10-fold higher selectivity toward Cdc25B phosphatase than Cdc25A [100].

Ethyl-(2*Z*)-5-(1,3-benzodioxol-5-yl)-2-(3,5-dibromo-4-hydroxybenzylidene)-7-methyl-3-oxo-2,3-dihydro-5*H*-[1,3]thiazolo [3,2-*a*]pyrimidine-6-carboxylate **97** is the most potent thiazolo-pyrimidine structure. It was synthesized and investigated by Braud et al., in 2009 and showed IC_50_ of 4.5 ± 0.2 µM on MBP–Cdc25B3 in a dose-dependent manner and 3.4 ± 1.8 µM on a LNCaP cell line. The mechanism of inhibition appeared to not be dependent on ROS release, which is contrary to the quinoid-based structure [101]. Moreover, the same research group modified the previous compound by application of azide-alkyne cycloaddition (click reaction) to obtain compound **98**. Compound **98** was a potent inhibitor with IC_50_ of 3.0 µM towards Cdc25B. Its activity included specific H-bond networking with the inhibitor binding pocket of the enzyme [102].

Ryu and his coworkers identified **99** and **100** as novel Cdc25 phosphatase inhibitors with micromolar activity utilizing a structure-based *de novo* design method. The IC_50_s of both structures toward Cdc25A and B were 5.1, 1.2, 4.7, 2.3 µM, respectively [103].

8-Hydroxy-7-(6-sulfonaphthalen-2-yl)diazenyl-quinoline-5-sulfonic acid (NSC-87877) **101** is an example of a competitive inhibitor with high selectivity against Cdc25A and Cdc25B with IC_50_ of 0.60 ± 0.004 and 1.29 ± 0.35 µM, respectively [104].

In 2010, Chen et al., delivered a series of *N*-substituted maleimide fused-pyrazole analogs with Cdc25B inhibitory activity, starting from a high-throughput screening hit. A simplified 3,5-diacyl pyrazole analog was introduced as a promising inhibitor (**102**, IC_50_ = 0.12 μM) with a 270-fold increase in potency [105].

A coumarine-based structure was first introduced as a potential Cdc25 inhibitor scaffold in 2010; **103** was the most active hit among other derivatives which were under investigation. The activity of **103** exhibited IC_50_ of 27 and 29 µM towards A and B isoforms, and it inhibited Cdc25A, B, and C by percentages of 94.2 ± 7.3, 35 ± 5.5, and 79.3 ± 2.8, respectively at 100 µM [106].

Using HTS techniques, Choi and his coworkers discovered a non-peptide, small inhibitor of PTP-1B that serves as an oral glucose-lowering agent. Among them, the KR61170 (B) **104** has shown PTP-1B inhibitory activity of IC_50_ equal to 0.29 ± 0.1, as well as Cdc25A, B inhibitory properties with IC_50_ of 1.5 ± 0.3 and 0.17 ± 0.1 µM, respectively [107].

In 2009, The University of Pittsburgh Molecular Library Screening Center (Pittsburgh, PA, USA) conducted a screen with the NIH compound library for inhibitors of Cdc25B. Six of the nonoxidative hits were selective for Cdc25B inhibition versus MKP-1 and MKP-3. Only the two bisfuran-containing hits, (**105**) and (**106**), significantly inhibited the Cdc25B with IC_50_’s of 11.6 and 15.5 μM, respectively. In addition, they significantly inhibited the growth of the MBA-MD-435 breast and PC-3 prostate cancer cell lines with IC_50_’s of 96.2 ± 68, 15.3 ± 1.36, 50.4 ± 8.6, and 47.5 ± 2.75 μM, respectively. They suppressed enzymes in a mechanism that appeared to be not related to H_2_O_2_ or ROS production, which makes it safer and interesting for further investigation [108].

(±)-3-(2-(2-Fluorobenzyloxy) naphthalen-6-yl)-2-aminopropanoic acid derivatives are structurally based on phosphotyrosine, and they were intentionally designed as PTP1B inhibitors. Among the studied compounds, **107** exhibited good activity toward Cdc25B (IC_50_: 2.49 ± 0.13 μM), with poor selectivity in comparison with other phosphatases PTP1B (IC_50_: 1.57 ± 0.22 μM) and TCPTP (IC_50_: 3.67 ± 0.46 μM) [109].

The well-known antitumor activity of coumarin and its diallyl polysulfides was the spark of the synthesis of new dicoumarine products linked together with di, tri, and tetra sulfide linkage. This group of derivatives exhibited activity against HCT116 cell lines in a time and dose-dependent manner. The reduction in cell viability was 30%, 22%, and 18% for **108**, **109**, and **110** when treated with 25 µM and to 50% in the concentration of 50 µM. ROS release was the proposed mechanism of cellular inhibition as it happened at the G2/M phase which is mainly regulated by Cdc25C phosphatase [110]. Compounds **108**, **109**, and **110** decreased the level of Cdc25A and C, and this activity was proportional to the increasing number of the sulfur atoms but less than the activity of BN82002 (**79**) [110].

An imidazopyridine derivative, 1-(3,4-dihydroxyphenyl)-2-((4-(furan-2-ylmethyl)-5-(p-tolyl)-4H-1,2,4-triazol-3-yl)thio) ethan-1-one (**111**, CHEQ-2), is structurally related to compound **108**. It demonstrated an improved inhibitory ability relative to XDW-1 and VK3 and showed favorable inhibition on the proliferation of tumor cells. Its IC_50_ against Cdc25 A, B, and HT-29, HepG2 cells lines were 0.39, 1.26, 24.20, and 11.25 μM, respectively. Similar to BN82685 **44**, CHEQ-2 (**111**) was bioavailable by the oral route of administration and halted the growth of xenografted human liver tumor growth in mice by 42.1% for 14 days of administration (10 mg/kg/day) with significant low toxicity (LD_50_ > 2000 mg/kg) [111].

Recently, the two triterpenoids 3β-hydroxylanosta-8,24*Z*-dien-26-oic acid (**112**) and cyclooctenol (**113**) were isolated from the stem and leaves of *Schisandra chinensis.* They showed significant inhibition of Cdc25A phosphatase with inhibitory rates of 85.5% and 98.1%, respectively at 100 µM [112].

Moreover, a series of 2,3-dioxoindolin-*N*-phenylacetamide derivatives was synthesized and evaluated for their inhibitory activity against Cdc25B and PTP1B enzymes. 2-(4-Chloro-2,3-dioxoindolin-1-yl)-*N*-phenylacetamide (**114**) showed the most inhibitory activity with IC_50_ values of 3.2 and 2.9 μg/mL against Cdc25B and PTP1B, respectively[113].

Early published work by Lauria et al., identified (**115**, J3955) as Cdc25 modulators by the correlation between the chemosensitivity of J3955 and the protein expression pattern of the target enzyme. J3955 (**115**) showed concentration-dependent antiproliferative activity against HepG2 cells with GI_50_ of 1.50 ± 0.37 µM. Western blotting analysis showed an increment of phosphorylated Cdk1 levels in cells exposed to J3955, indicating its specific influence in cellular pathways involving Cdc25 A, B, and C proteins with IC_50_’s of 1.12 ± 0.09, 2.19 ± 0.07, and 2.22 ± 0.07 µM, respectively [114]. The mentioned miscellaneous Cdc25 inhibitors **70**–**115** are represented in Figure 9.

## 8. Computational Methods to Discover and Optimize Cdc25 Inhibitors

Molecular modeling and other computational techniques have been applied for screening of large libraries of compounds or for much greater comprehension of the mode of interaction of already proven active compounds.

One of the earlier molecular modeling computational analyses was applied for vitamin K_3_ derivative (Cpd5,**2**) with Cdc25A phosphatase. Cpd 5 (**2**) was found to be closer to the enzyme active site than VK_3_. It was bound with more favorable energy forming an adduct with the catalytic cysteine [25].

In 2006, Lavecchia et al., investigated the structural models for the interaction of various Cdc25B inhibitors namely **16** and **51** with the enzyme via AUTODOCK and GOLD computational docking programs [115]. In GOLD model, compound **51** was found to have dihydroxyquinone ring fitting in catalytic sites and 7-(2-methyl-benzyl)indolyl was placed between Arg482 and Arg544. The 4-carbonyl oxygen accepted the H-bonds from Arg479, and the 5-OH was involved in H-bonding with Phe475, Ser477, and Glu478. For compound **16** the more likely pose (AUTODOCK model) involved electrostatic interaction between 5,8-oxygen of the quinone and Arg544, Arg482, and Tyr428. The nitrogen atom of pyridine made a hydrogen bond with Arg482. The oxygen of the side-chain shared in this hydrogen bonds network with Phe475 and Arg479 (Figure 10) [115].

In 2008, Park and his coworkers studied the mode of the interaction of NSC 95397 (**8**) by AUTODOCK and nanosecond molecular dynamics simulations. The molecule was fitted into the active site of Cdc25B forming hydrogen bonds with Glu478, Glu474, Arg544, and Tyr428. It was also found that the hydrophobic interaction plays a critical role in the strong stabilization of NSC 95397 (**8**) [116].

Another structure-Based virtual screening trial based on molecular modeling led to the discovery of some structures. Among them, **116** was the most potent with IC_50_ of 0.82 ± 0.41 µM (Cdc25A) and 1.98 ± 0.44 µM (Cdc25B). The binding mode of **118** was determined using the available crystal structure of Cdc25B and a homology modeling of the Cdc25A structure. The orientation of the molecule in both enzymes was different due to the smaller active site of Cdc25A. Dihydroxyphenylcarbonyl moiety formed hydrogen bonds with the backbone of the enzyme while triazole moiety mimics the phosphate group of the substrate. The toluene group was stabilized by hydrophobic interaction with Trp507 and Arg501 in the Cdc25A which may explain the increased activity against it (Figure 11) [117].

Another trial of structure-based virtual ligand screening computations was done with FRED, Surflex, and LigandFit, towards a compound collection of over 310,000 druglike molecules and the crystal structure of Cdc25B. Ninety-nine compounds were identified as promising Cdc25 inhibitors that also exhibited antiproliferative properties. In the same study, the biological analysis of **117** and **118** (Figure 12) proved them as the most potent among the tested molecules (IC_50_ of 13.0 ± 0.5 and 19 ± 1.3 µM) toward Cdc25B and (IC_50_ of 15.8 ± 1.8 and 3.6 ± 1.2 µM) toward HeLa cells, respectively. The results previously obtained according to the bioassay did not match the computational analysis [118].

Using de novo drug design probed by LigBuilder program, structure **119** was identified (Figure 12). It was tested on Cdc25A and B to give IC_50_ equal to 2.3 and 1.7 µM, respectively. The mode of interaction with the enzyme was examined by AUTODOCK. It was found to be fitted in a slightly different way between Cdc25A and B. The terminal phenyl ring was inserted deeper in the larger active site of the B isoform and established hydrophobic interaction with Tyr428, Arg479, and Met531 which may describe the higher activity against this isoform. Cyano and furan moieties interact with both enzymes in a similar pattern of hydrogen bonds while the thiol group appeared to surrogate the phosphate of the substrate in both active sites [119].

A molecular field point strategy was recently employed by a Field Templater software package to scan the pharmacophores which are essential for the activity of reversible Cdc25s inhibition. Three derivatives of scaffolds thiazolopyrimidine, pyrazole, and naphthofurandione, which are known as reversible inhibitors, were used as templates to build up a pharmacophore model. They were used for scanning a wide library of compounds (~3.5 million) and introducing promising candidates which would be subjected to biological studies. Compound **120** (Figure 12) was an example of these products offered by this type of drug discovery. It exhibited IC_50_’s of 2.7 ± 1.7, 1.7 ± 2.0, and 9.7 against A, B, and C isoforms, respectively, and 26.0 ± 1.0 on VHR [120].

In 2019, Ferrari and his co-workers conducted a pharmacophore-guided drug discovery program for the identification of scaffolds of the naphthoquinone group displaying inhibition of Cdc25. They found that 2-(2′,4′-dihydroxyphenyl)-8-hydroxy-1,4-naphthoquinone (**121**, UPD-140) and 5-hydroxy-2-(2,4-dihydroxyphenyl) naphthalene-1,4-dione (**122**, UPD-176) were the most potent compounds that induced inhibition of CDK1 activity and blocked the mitotic transition followed by cell death (Figure 13). In mouse Apc/K-Ras mutant duodenal organoids, low doses of Cdc25 inhibitors caused the arrest of proliferation and expression of differentiation markers, whereas high doses induced cell death. In zebrafish embryos, used as in vivo xenograft models, the Cdc25 inhibitors led to tumor regression and reduction in metastases [121].

## 9. Conclusions

This review aims to summarize the present state-of-the-art regarding design of Cdc25 phosphatase inhibitors. The discovery of new Cdc25 inhibitors is extremely significant for cancer therapy due to the critical roles that these phosphatases play in regulating cell-cycle progression. From the studies mentioned in this review article, no certain scaffold can be identified as an exclusive Cdc25 phosphatase inhibitor. Many potent Cdc25 inhibitors were synthesized or isolated from natural extracts. Meanwhile, quinoids are considered the most potent inhibitors despite their toxicity. Quinoids act mainly as Michael acceptors for the negative charge of Cys-S^−^ in the catalytic domain. It seems that the small size of quinoids allows them to be suited to the adjacent swimming pool which gives them a good chance to play their roles correctly. One of the promising perspectives for finding an efficient inhibitor may depend on the optimization of the already discovered quinoids to eliminate their toxicity without diminishing their activity. Molecular modeling and other computational techniques have been applied for screening large libraries of compounds and to shed the light on the mode of interaction of already proven active compounds.

## Figures and Tables

**Figure 1 molecules-27-02389-f001:**
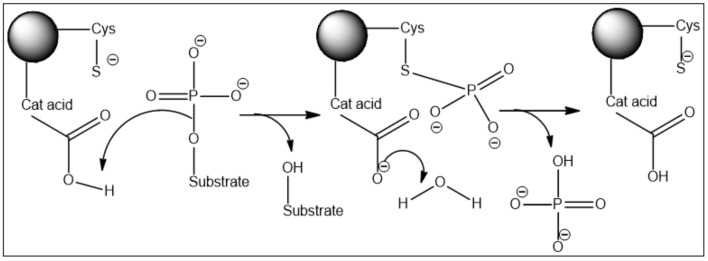
Two-step reaction mechanism of the Cdc25 phosphatases with a covalent phospho-cysteine intermediate.

**Figure 2 molecules-27-02389-f002:**
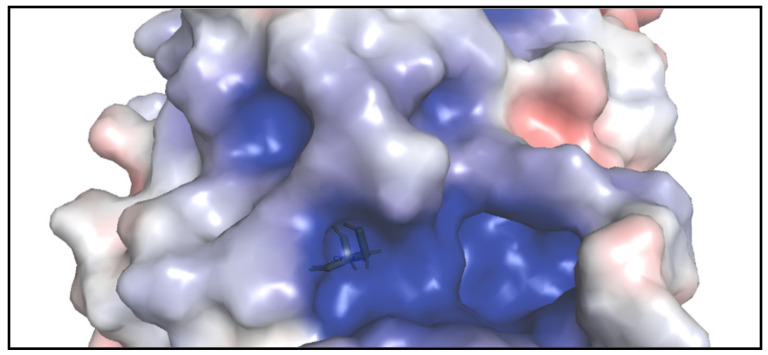
The surface of Cdc25B shows the shallow active site (where the stick of Cys473 is shown) and the adjacent inhibitory pocket which is deeper and larger.

**Figure 3 molecules-27-02389-f003:**
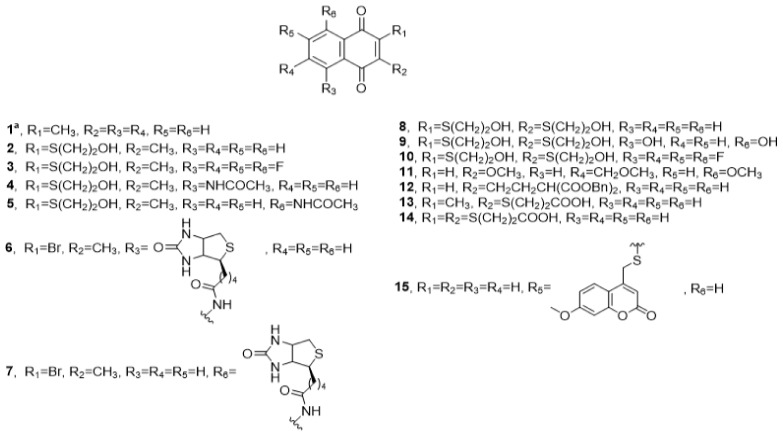
Cdc25 inhibitors based on quinoid structure.

**Figure 4 molecules-27-02389-f004:**
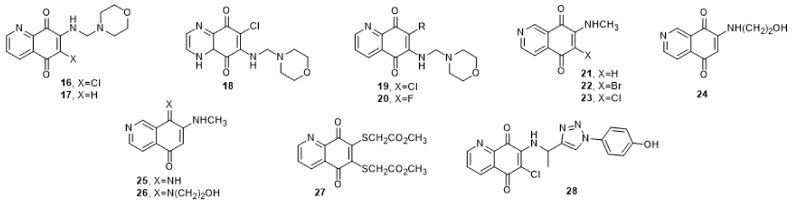
Quinone structural analogs of vitamin K_3_ substituted with heteroatoms in the benzene ring.

**Figure 5 molecules-27-02389-f005:**
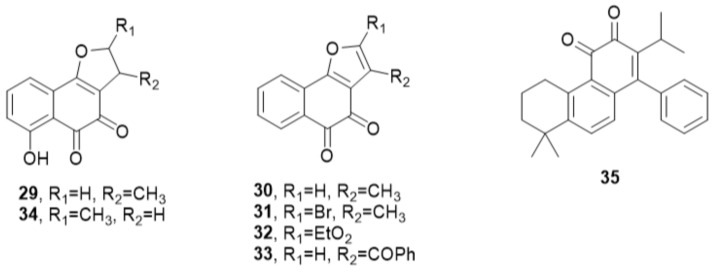
New quinone structures.

**Figure 6 molecules-27-02389-f006:**
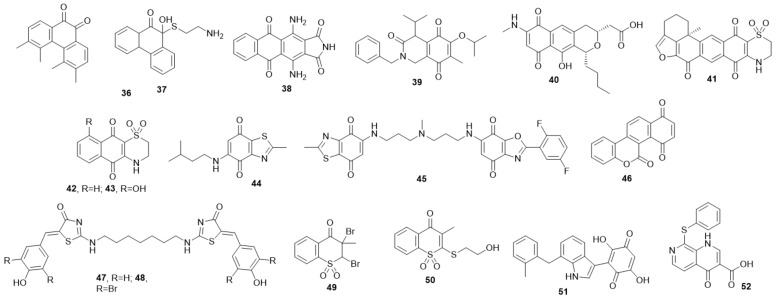
A library of small molecules structurally related to paraquinoid and non-para quinoid to be Cdc25 dual specificity phosphatase inhibitors.

**Figure 7 molecules-27-02389-f007:**
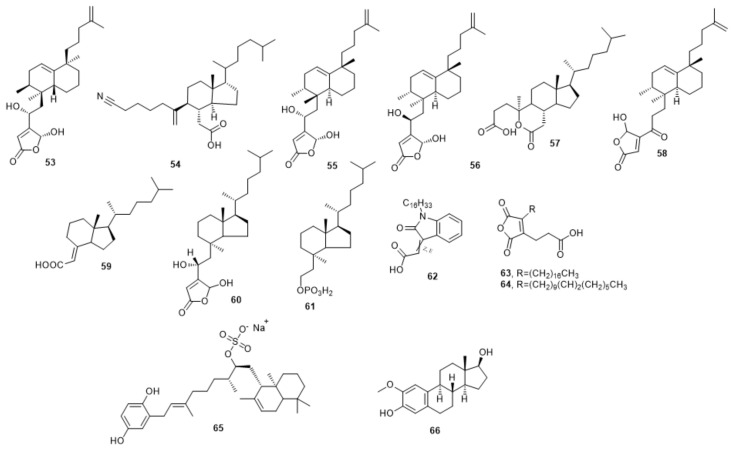
Cdc25 inhibitors based on steroid- and dysidolide-like compounds.

**Figure 8 molecules-27-02389-f008:**
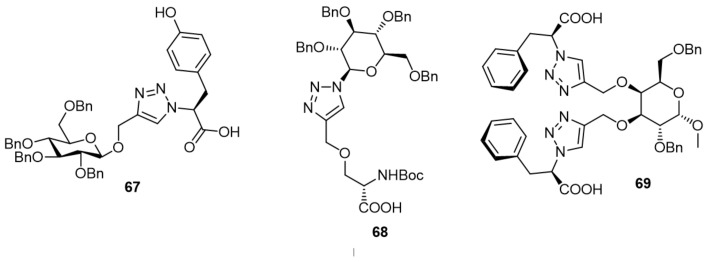
Cdc25 inhibitors based on Glycosidic.

**Figure 9 molecules-27-02389-f009:**
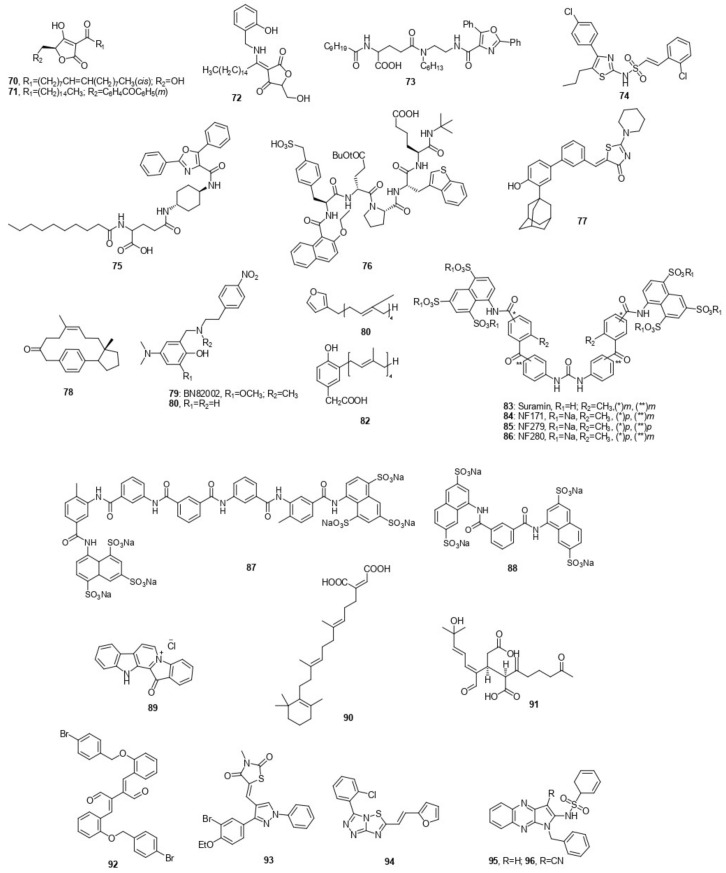
Miscellaneous Cdc25 inhibitors.

**Figure 10 molecules-27-02389-f010:**
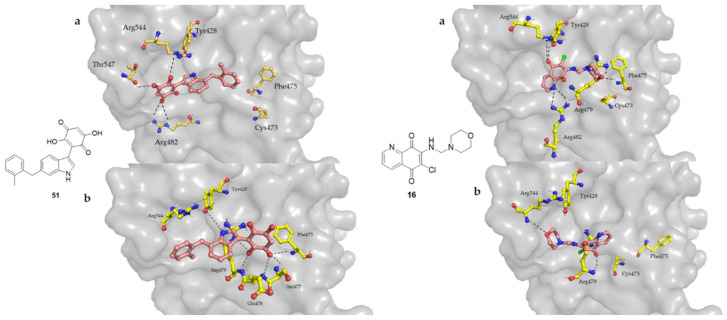
The binding modes of compounds **51** and **16** inside the binding groove of Cdc25B simulated by **a** AUTODOCK and **b** GOLD.

**Figure 11 molecules-27-02389-f011:**
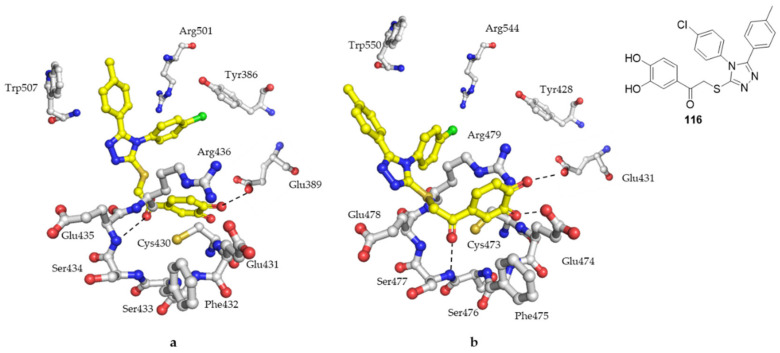
The theoretical binding orientations of **116** (yellow stick) in the active sites of (**a**) Cdc25A and (**b**) Cdc25B.

**Figure 12 molecules-27-02389-f012:**
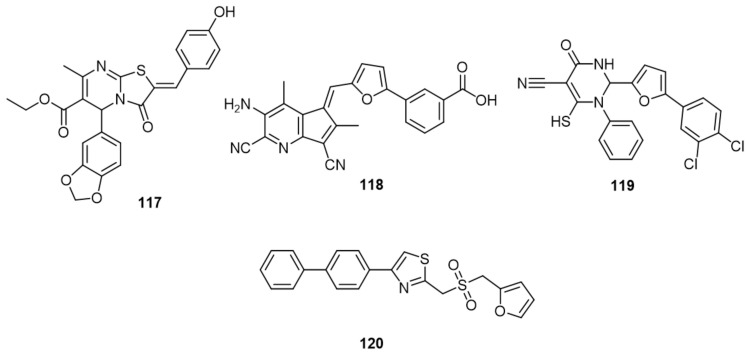
Chemical structures of compounds **117**, **118**, **119** and **120**.

**Figure 13 molecules-27-02389-f013:**
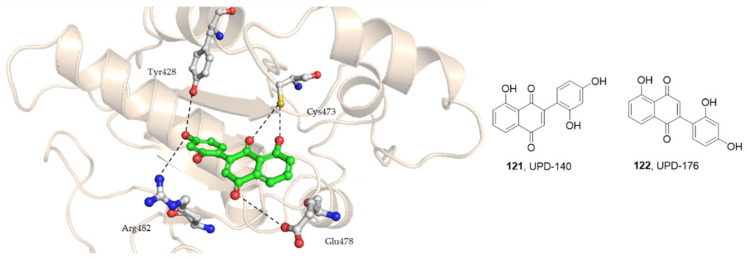
The chemical structures of **121**, **122**, and the docking of UPD-140, **121** into the Cdc25B catalytic site.

**Table 1 molecules-27-02389-t001:** Cdc25 inhibitors based on quinoid structure.

ID	Cdc25 Inhibitory			Hep-3B	MDA-MB-231	HTERT-HME1	HeLa	HepG2	PLC/PRL/5	MCF-7	FemX	LS-140	SKBR3	HR	A549
	A	B	C												
**1** [10,11]	38 ± 4	95 ± 3	20 ± 4												
**2** [26,27]	2	2	2	6.3 ± 0.4				15 ± 1	12 ± 2	12.9 ± 0.3					
**3** [27]				2				1			1	10	15	15	
**4** [28]	1.5														
**5** [28]	1.5														
**6** [29]	1.5			20											
**7** [29]	1.5			20											
**8** [30]	0.022 ± 0.0059	0.125 ± 0.006	0.0569 ± 0.0177	6 ± 0.9			10.8 ± 2.9	7 ± 0.5	6 ± 1	10.3 ± 0.4					7.59 ± 0.44
**9** [26]				1.2 ± 0.2				1.5 ± 0.5	1.3 ± 0.2	1.8 ± 0.5					
**10** [27]															
**11** [31]	0.4														
**12** [32]		10.7 ± 0.5		12.2 ± 3.7											
**13** [32]		4.55 ± 0.16		51.3 ± 4.1											
**14** [33]		1/17 ± 0.04													
**15** [34]		9.1 ± 0.7	9.1 ± 0.7	0.9 ± 0		9.5 ± 1.0	18.1 ± 1.3				11.2 ± 0.7				
		45.5 ± 8 C-cata	45.5 ± 8 C-cata												

All the IC_50_ values expressed as µM.

**Table 3 molecules-27-02389-t003:** New quinoid structures based on tanshinone patterns as Cdc25 inhibitors.

ID	Cdc25 Inhibitory			A549	HeLa	SBC-5	U937	PC-3	MDA-MB-435	HCT-116
	A	B	C							
**29** [43]		3.21 ± 0.76		2.07						
**30** [44]	0.73 ± 0.1	0.65 ± 0.09		1.66 ± 0.13						
**31** [44]	0.73 ± 0.04	0.48 ± 0.03		3.14 ± 0.13						
**32** [44]	1.10 ± 0.11	0.89 ± 0.07		2.80 ± 0.2						
**33** [45]	5.0 ± 0.1	10.4 ± 0.1	8.8 ± 0.1					6.5	1.2	
**34** [46]		17			0.38	0.54	6.25			
**35** [47]	3.96 ± 0.63	3.16 ± 0.22	4.11 ± 0.40	6.24 ± 0.17						6.26 ± 0.12

All IC_50_ values measured by µM.

**Table 4 molecules-27-02389-t004:** Cdc25 inhibitors based on steroid and dysidolide-like compounds.

ID	Cdc25 Inhibitory			A-549	P388	SBC-5	HL60	PLC\PRF5	Hep3B	McH7777	JM-1	HepG2
	A	B	C									
**53** [62,63,64]	9.4, 35	87		4.7	1.5	5.4	7.1					
**54** [62,65,66]	2.2											
**55** [64]	15	19				16	13.1					
**56** [64]	13	18				1.3	1.0					
**57** [67]	7.9											
**58** [68]	0.8											
**59** [69]	7.7					47						
**60** [70,71]	0.48	1.44										
**61** [70,71]	0.7	3.7										
**62** [72]	1.9 ± 1.0											
	1.6 ± 0.2											
**63** [73,74]	5.1 ± 0.3	3.6 ± 0.4	4.3 ± 1.1									
**64** [73,74]	4.5 ± 1.7	6.7 ± 0.6	8.8 ± 1.3									
**65** [75]	3											
**66** [76]	1	1	10					0.5	1	1	1	3

All IC_50_ values measured by µM.

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
