# Peer review of "A Comprehensive Overview of the Developments of Cdc25 Phosphatase Inhibitors"

_molecules, 2022, doi:10.3390/molecules27082389_

Round 1
Reviewer 1 Report
The review by Ahmed Bakr Abdelwahab et al. presents the state-of-the-art in the field of Cdc25 modulators. My personal expertise is biology, so I am not in position of evaluating chemical parts.
In general, the study is a detailed and comprehensive work that deserves publication.
However, the text needs a thorough editing. The sentence 'When it is tested with hydrogen peroxide.....' (lines 74-76) is unclear. Line 86: primary human cancers - how about metastatic and recurrent tumors? Line 371: based on glycosidic - what? is the word missing?
The section 3 Conclusion does not contain an in-depth analysis of the material. The study would win if the authors provide their opinion regarding controversies, limitations and advantages of Cdc25 targeting agents.
The editing should include both linguistic and scientific corrections across the entire manuscript.
Author Response
"Please see the attachment."

Reviewer 2 Report
In the introduction, you have to include also the role of inhibitors, not only the isoforms.
Line 73: remove the semicolon
Line 154: Cdc35 is a mistake? It is Cdc25?
Why in the tables the authors reported the data in uM/ml and not as nM or uM concentrations as in the text? Also I suggest to include the reference in a column for each compound described in the table to facilitate the reader.
I think that a paragraph describing the principal roles in cancer research must be included.
Author Response
"Please see the attachment."

Round 2
Reviewer 1 Report
The authors revised the manuscript. Still, the quality of the English language remains to be improved. This can be done by the professional editor.
To me the fragment about vitamin K (lines 45-49) is not on the right place. Chemical classes of Cdc25 inhibitors are described in detail below, so I guess there is no necessity to refer to vitamin K and quinoids in Introduction.